# MiR-497∼195 cluster regulates angiogenesis during coupling with osteogenesis by maintaining endothelial Notch and HIF-1α activity

Mi Yang[1,2,*], Chang-Jun Li[1,2,*], Xi Sun[1,3], Qi Guo[1,4], Ye Xiao[1], Tian Su[1], Man-Li Tu[1,2], Hui Peng[1,3], Qiong Lu[1], Qing Liu[5], Hong-Bo He[5], Tie-Jian Jiang[1], Min-Xiang Lei[1], Mei Wan[2], Xu Cao[2] & Xiang-Hang Luo[1]

A specific bone vessel subtype, strongly positive for CD31 and endomucin (CD31$^{hi}$Emcn$^{hi}$), is identified as coupling angiogenesis and osteogenesis. The abundance of type CD31$^{hi}$Emcn$^{hi}$ vessels decrease during ageing. Here we show that expression of the miR-497∼195 cluster is high in CD31$^{hi}$Emcn$^{hi}$ endothelium but gradually decreases during ageing. Mice with depletion of miR-497∼195 in endothelial cells show fewer CD31$^{hi}$Emcn$^{hi}$ vessels and lower bone mass. Conversely, transgenic overexpression of miR-497∼195 in murine endothelium alleviates age-related reduction of type CD31$^{hi}$Emcn$^{hi}$ vessels and bone loss. miR-497∼195 cluster maintains the endothelial Notch activity and HIF-1α stability via targeting F-box and WD-40 domain protein (Fbxw7) and Prolyl 4-hydroxylase possessing a transmembrane domain (P4HTM) respectively. Notably, endothelialium-specific activation of miR-195 by intravenous injection of aptamer-agomiR-195 stimulates CD31$^{hi}$Emcn$^{hi}$ vessel and bone formation in aged mice. Together, our study indicates that miR-497∼195 regulates angiogenesis coupled with osteogenesis and may represent a potential therapeutic target for age-related osteoporosis.

[1] Department of Endocrinology, Endocrinology Research Center, Xiangya Hospital of Central South University, Changsha, Hunan 410008, China. [2] Department of Orthopaedic Surgery, Johns Hopkins University School of Medicine, Baltimore, Maryland 21205, USA. [3] Department of Endocrinology, The Second Xiangya Hospital of Central South University, Changsha, Hunan 410011, China. [4] Key Laboratory of Organ Injury, Aging and Regenerative Medicine of Hunan Province, Changsha 410008, China. [5] Department of Orthopedic Surgery, Xiangya Hospital, Central South University, Changsha 410008, China. * These authors contributed equally to this work. Correspondence and requests for materials should be addressed to X.C. (email: xcao11@jhmi.edu) or to X.-H.L. (email: xianghangluo@hotmail.com).

Different tissues have distinct metabolic requirements. Therefore, specific vessels are formed in specific tissue to provide molecular signal which decides the fate of progenitor cells and mediates cell differentiation[1–4]. A specific endothelium was identified in murine skeletal system that is strongly positive for CD31 and endomucin with signalling properties that support bone formation and regeneration[5,6]. This new capillary type is named CD31$^{hi}$Emcn$^{hi}$ or type H vessels and able to mediate perivascular osteoprogenitors differentiation and couple angiogenesis to osteogenesis[5]. The number of type H endothelial cells decreases during ageing correlating with the age-associate bone loss. However, the underlying mechanism of type H vessels formation and degeneration remains elusive.

MicroRNAs (miRNAs) are a class of small ($\sim$22 nucleotides), single-stranded noncoding RNAs found in diverse organisms, which regulate the expression of mRNAs by binding the 3′ untranslated regions (UTRs) or amino acid coding sequences[7–9]. Post-transcriptional regulation mediated by miRNAs is thought to be one of the most important cellular regulation mode. Studies have suggested that miRNAs play crucial roles in various aspects of bone metabolism[10–12]. However, the roles of miRNAs in the regulation of type H vessels biology is still unclear.

In the present study, we show that miR-497$\sim$195 cluster is highly expressed in CD31$^{hi}$Emcn$^{hi}$ endothelial cells and contributes to the formation of CD31$^{hi}$Emcn$^{hi}$ vessels. We find that miR-497$\sim$195 promotes the formation of CD31$^{hi}$Emcn$^{hi}$ endothelium by maintaining endothelial Notch and HIF-1$\alpha$ activity. Thus, our study provides an important mechanism of miRNA regulating angiogenesis during coupling with osteogenesis.

## Results

**MiR-497$\sim$195 cluster was strongly expressed in CD31$^{hi}$Emcn$^{hi}$ endothelial cells.** To determine the role of miRNAs in type H vessel generation, CD31$^{hi}$Emcn$^{hi}$ and CD31$^{lo}$Emcn$^{lo}$ endothelial cells were sorted by fluorescence-activated cell sorting (FACS) from bone marrow cells of 1-month-old C57BL/6 mice to identify dysregulated miRNAs by performing miRNA microarray analysis. Among them, the expression of miR-497 and miR-195, which are found clustered at the same locus, is 3 times higher in CD31$^{hi}$Emcn$^{hi}$ than in CD31$^{lo}$Emcn$^{lo}$ endothelial cells (Fig. 1a). The higher level of miR-497$\sim$195 cluster expression in CD31$^{hi}$Emcn$^{hi}$ endothelial cells was further confirmed by quantitative real-time PCR (qRT-PCR; Fig. 1b). MiR-497 and miR-195 belong to miR-15 family, thus we also tested the expression of other members of the miR-15 family in bone marrow endothelial cells (BMECs). We observed moderate elevation of miR-15a, miR-15b and miR-16 expression levels, and great increase in miR-497$\sim$195 expression in CD31$^{hi}$Emcn$^{hi}$ endothelial cells compared to CD31$^{lo}$Emcn$^{lo}$ endothelial cells (Supplementary Fig. 1a). Notably, the miR-497$\sim$195 levels in human and mice endothelial cells sorted from bone marrow cells were negatively correlated with age (Fig. 1c–f). As the animals aged, the decreasing of the miR-497$\sim$195 cluster in endothelial cells was correlated with the pronounced reduction of CD31$^{hi}$Emcn$^{hi}$ vessels and remarkable decrease of Osterix$^{+}$ osteoprogenitors (Fig. 2a–d) as well as the loss of bone mass (Fig. 2e–i). These data suggest that miR-497$\sim$195 may play an important role in regulating type H vessel formation and degeneration along with ageing in mouse and human.

**MiR-497$\sim$195 cluster knockout mice showed impaired CD31$^{hi}$Emcn$^{hi}$ vessels and bone formation.** To investigate the role of miR-497$\sim$195 in endothelial cells *in vivo*, we generated endothelial-cell-specific miR-497$\sim$195 cluster knockout (*miR-497$\sim$195$^{-/-}$*) mice by combining loxP-flanked miR-497$\sim$195 alleles (*miR-497$\sim$195$^{lox/lox}$*) and Cdh5 (PAC)-Cre transgene. qRT-PCR confirmed that the expression level of miR-497$\sim$195 cluster in *miR-497$\sim$195$^{-/-}$* mice was one third of which in *miR-497$\sim$195$^{lox/lox}$* controls (Supplementary Fig. 2a). The expression of miR-15a, miR-15b and miR-16 showed no significant difference between *miR-497$\sim$195$^{-/-}$* and *miR-497$\sim$195$^{lox/lox}$* controls indicating that genetic manipulation of miR-497$\sim$195 in ECs has little effect on other members of the miR-15 family (Supplementary Fig. 2b).

Co-immunostaining of CD31 and Emcn demonstrated that the amount of CD31$^{hi}$Emcn$^{hi}$ endothelium in the bone was strongly reduced in *miR-497$\sim$195$^{-/-}$* mice compared to their *miR-497$\sim$195$^{lox/lox}$* littermates (Fig. 3a). Flow cytometric analysis showed a significant decrease of CD31$^{hi}$Emcn$^{hi}$ endothelial cells number in bone marrow of *miR-497$\sim$195$^{-/-}$* mice as well as decreased fraction of EdU$^{+}$ bone ECs (Fig. 3c,d and Supplementary Fig. 2c,d). However the total endothelial cell number and the branch number of $\alpha$-SMA$^{+}$ artery were not significantly changed in *miR-497$\sim$195$^{-/-}$* mice (Fig. 3e and Supplementary Fig. 2e,f). Microcomputed tomography ($\mu$-CT) showed significantly lower trabecular bone volume, number, thickness and higher trabecular separation in *miR-497$\sim$195$^{-/-}$* mice compared to *miR-497$\sim$195$^{lox/lox}$* controls (Fig. 4a–e). Calcein double labelling confirmed that *miR-497$\sim$195$^{-/-}$* mice had decreased bone formation rates (BFRs; Fig. 4f–h). *miR-497$\sim$195$^{-/-}$* mice also showed significantly decreased number of Osterix$^{+}$ osteoprogenitors and bone surface osteoblasts relative to their control littermates (Fig. 3a,b and Supplementary Fig. 3a,b), however, the number and surface of osteoclasts in bone surface were unchanged (Supplementary Fig. 3c–e). Taken together, these results suggest that *miR-497$\sim$195$^{-/-}$* mice have reduced CD31$^{hi}$Emcn$^{hi}$ vessels and bone formation.

**MiR-497$\sim$195 cluster transgenic mice showed increased CD31$^{hi}$Emcn$^{hi}$ vessels and bone formation.** We then generated transgenic mice overexpressing miR-497$\sim$195 specifically in endothelial cells to investigate whether overexpression of miR-497$\sim$195 *in vivo* would contribute to CD31$^{hi}$Emcn$^{hi}$ vessels growth and bone formation. qRT-PCR revealed five folds higher of miR-497$\sim$195 expression in Tg mice as compared with that in controls (Supplementary Fig. 4a) and no alterations in the transcription of miR-15a, miR-15b and miR-16 (Supplementary Fig. 4b).

Tg mice had pronouncedly increased number of CD31$^{hi}$Emcn$^{hi}$ endothelial cells compared to wild-type (WT) mice detected by immunofluorescence staining and FACS analysis (Fig. 5a,c–e). The fraction of EdU$^{+}$ ECs in bone marrow was significantly increased in Tg mice relative to their WT control indicating a positive regulatory role for miR-497$\sim$195 in bone ECs proliferation. However, the branch number of the $\alpha$-SMA$^{+}$ arteries in Tg mice did not differ from the WT mice, as judged by $\alpha$-SMA immunofluorescent staining (Supplementary Fig. 4c–f). Tg mice were also characterized by an increased number of Osterix$^{+}$ osteoprogenitors and osteoblasts (Fig. 5a,b and Supplementary Fig. 5a,b), increased bone mass (Fig. 5f–j) and a higher BFR relative to controls (Fig. 5k,l). The number and surface of TRAP$^{+}$ osteoclasts were not affected in Tg mice (Supplementary Fig. 5c–e). Taken together, the results suggest that overexpression of miR-497$\sim$195 in endothelial cells promotes CD31$^{hi}$Emcn$^{hi}$ vessels formation and osteogenesis in mice.

**MiR-497$\sim$195 cluster maintained Notch and HIF-1$\alpha$ activity in endothelial cells.** BMECs were sorted by FACS and transfected with the agomiR-497$\sim$195 or antagomiR-497$\sim$195 to

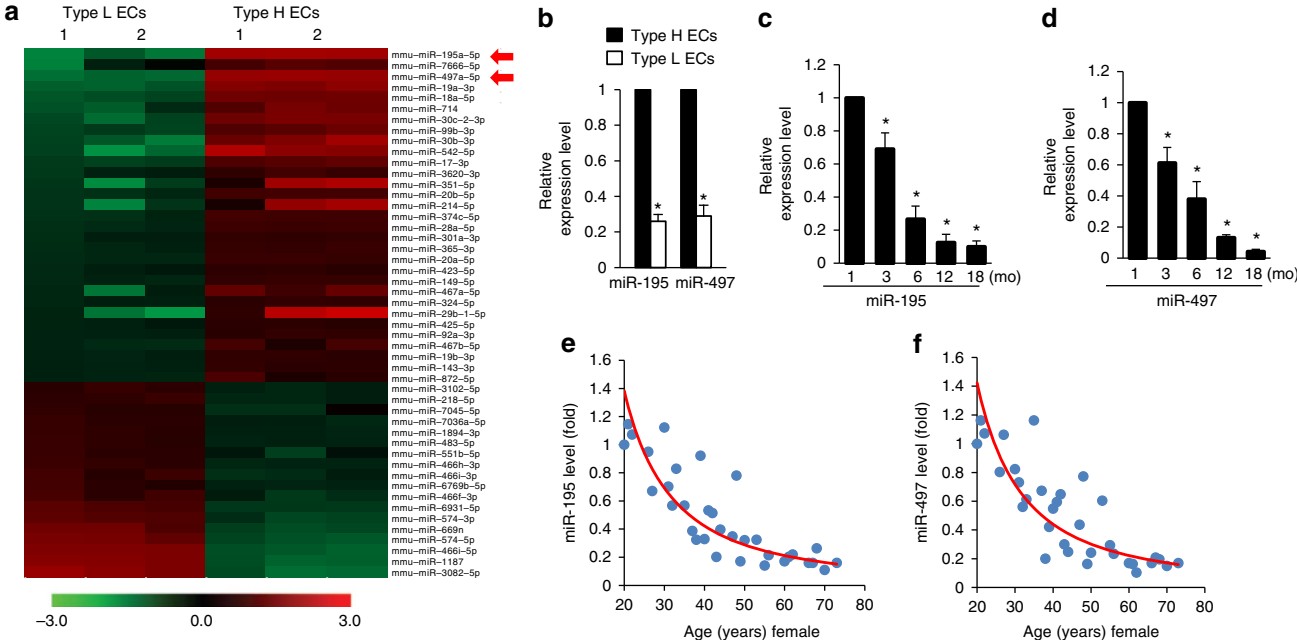

**Figure 1 | MiR-497~195 cluster was strongly expressed in CD31$^{hi}$Emcn$^{hi}$ endothelial cells.** (**a**) Microarray profiling results of deregulated miRNAs in CD31$^{hi}$Emcn$^{hi}$ and CD31$^{lo}$Emcn$^{lo}$ endothelial cells (Type H ECs and Type L ECs). ($n=3$ per group). (**b**) qRT-PCR analysis of miR-195/miR-497 level in Type H ECs and Type L ECs. ($n=5$ in each group from three independent experiments). (**c,d**) qRT-PCR analysis of the levels of miR-195 (**c**) and miR-497 (**d**) expression in endothelial cells derived from the mice at different ages. ($n=5$ mice in each group from three independent experiments). (**e,f**) Age-associated changes of miR-195 (**e**) and miR-497 (**f**) levels in endothelial cells from 33 human females. Data shown as mean ± s.d. *$P<0.05$, (**b**, Student's $t$-test; (**c,d**), analysis of variance (ANOVA)).

overexpress or silence miR-497~195 cluster, respectively (Fig. 6a,b). The BMECs were then cultured in endothelial cell growth medium. Overexpression of miR-497~195 promoted the CD31 and endomucin mRNA expression (Fig. 6c,d), whereas inhibition of miR-497~195 attenuated the CD31 and endomucin mRNA expression in BMECs (Fig. 6c,d).

Notch signalling and HIF-1α (hypoxia inducible factor 1, alpha subunit) have been reported to be involved in the generation of type H vessel[5,6]. The number of type H endothelium was dramatically reduced in mice with endothelial-cell-specific disruption of notch signalling. Endothelial-cell-specific over-expression of the Notch1 intracellular domain (NICD) increased the number of type H vessel[6]. We found that HIF-1α and NICD protein levels were up-regulated by overexpression of miR-497~195 (Fig. 6e–g), however the mRNA level of HIF-1α and Notch1 were not changed (Supplementary Fig. 6a,b). By contrast, inhibition of miR-497~195 attenuated NICD and HIF-1α protein in BMES without influencing the mRNA level of Notch1 and HIF-1α. (Fig. 6e–g and Supplementary Fig. 6a,b).

These results suggested that miR-497~195 promote the formation of CD31$^{hi}$Emcn$^{hi}$ endothelial cells by maintaining endothelial Notch and HIF-1α activity.

**MiR-497~195 cluster targets Fbxw7 and P4HTM.** To predict the targets of miR-497~195, we employed TargetScan. Among the predicted genes, Fbxw7 (F-box and WD-40 domain protein 7) and P4HTM (Prolyl 4-hydroxylase, transmembrane) genes were the novel targets of miR-497~195. Fbxw7 mediates the poly-ubiquitination and proteasomal degradation of active Notch[13], whereas P4HTM mediates the degradation of HIF-1α (refs 14,15).

Our results demonstrated that overexpression or inhibition of miR-497~195 cluster altered endogenous levels of Fbxw7 and P4HTM protein without changing mRNA level (Fig. 6e,h,i and Supplementary Fig. 6c,d). P4HTM belongs to prolyl hydroxylase domain proteins (PHDs) family. We also detected the expression

of other PHDs (PHD1, PHD2 and PHD3) in BMECs. However, the levels of PHD1, PHD2 and PHD3 protein in BMECs were unaffected by antagomiR-497~195 or agomiR-497~195 treatment (Supplementary Fig. 6e,f). Sequence analysis revealed conserved binding site for miR-497~195 cluster in the 3'-UTR of the Fbxw7 and P4HTM gene (Supplementary Fig. 6g). To clarify whether miR-497~195 cluster can directly regulate Fbxw7, luciferase reporter constructs containing the unaltered or mutated predicted miRNA-binding site of Fbxw7 (WT-pGL3-Fbxw7 and MUT-pGL3-Fbxw7, respectively) were generated. We transfected WT-pGL3-Fbxw7 or MUT-pGL3-Fbxw7 with agomiR-497~195 or agomiR-NC into BMECs and measured the effects of miR-497~195 on luciferase translation by the level of luciferase enzyme activity. AgomiR-497~195 repressed the luciferase activity of the Fbxw7 3'-UTR reporter gene, while MUT-pGL3-Fbxw7 abolished this inhibition (Fig. 6j). The luciferase reporter experiment for P4HTM also showed similar results as for Fbxw7 (Fig. 6k). These data show that miR-497~195 bind specifically to the 3'-UTR of Fbxw7 or P4HTM.

Small interfering RNA (siRNA) plays role in the RNA interference pathway by causing mRNA to break down after transcription[16]. We transfected cultured BMECs with Fbxw7 siRNA to silence Fbxw7 gene. The western blot analysis showed significantly lower Fbxw7 but higher NICD protein levels after transfection of siRNA-Fbxw7 compared to the control group (Fig. 6l,m). Using the same strategy, BMECs transfected with P4HTM siRNA showed significant increase of HIF-1α protein level (Fig. 6n,o).

These results demonstrate that miR-497~195 maintains Notch and HIF-1α activity by targeting Fbxw7 and P4HTM.

**Injection of aptamer-agomiR-195 increased CD31$^{hi}$Emcn$^{hi}$ vessel and bone formation.** To investigate the therapeutic effects of EC-specific overexpression of miRNA on age-related osteo-porosis, we constructed EC-specific aptamer with defined

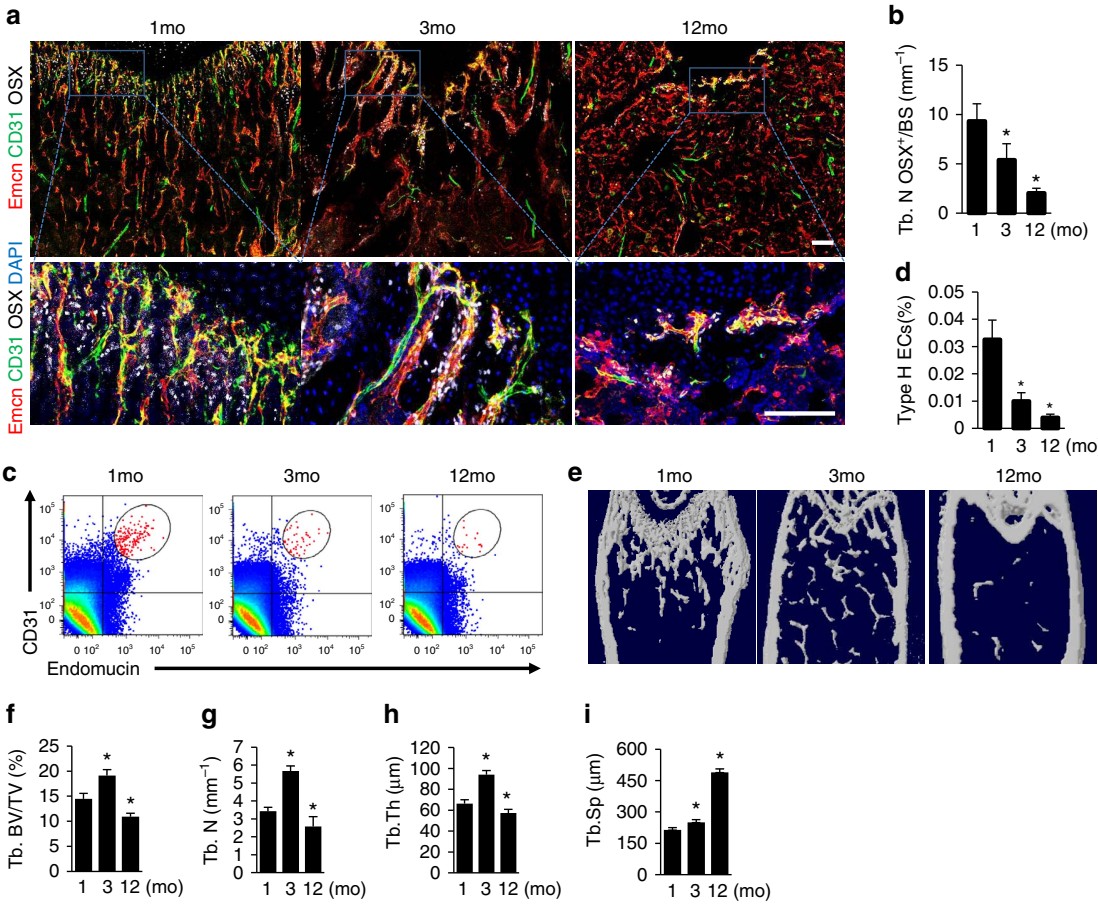

**Figure 2 | CD31$^{hi}$Emcn$^{hi}$ vessels and bone formation decreased during ageing.** (**a**) Representative images of CD31 (green), Emcn (red) and Osterix (white) immunostaining. Nuclei, DAPI (blue). Scale bar, 100 μm. (**b**) Quantitative analysis of Osterix$^{+}$ osteoprogenitors in distal femora. ($n = 5$ mice in each group from three independent experiments). (**c**,**d**) FACS analysis dot plot (**c**) and quantitation of Type H ECs (**d**) from long bone of the indicated age group. ($n = 6$ mice in each group from three independent experiments). (**e–i**) Representative microcomputed tomography (μCT) images (**e**) and quantitative μCT analysis (**f–i**) of trabecular bone microarchitecture in femora from 1-, 3- and 12-month-old WT mice. (Tb. BV/TV, trabecular bone volume per tissue volume; Tb.N, trabecular number; Tb.Th, trabecular thickness; Tb.Sp, trabecular separation. $n = 5$ mice in each group from three independent experiments). Data shown as mean ± s.d. *$P < 0.05$, (analysis of variance (ANOVA)).

structures which could specifically target endothelial cells using systematic evolution of ligands by exponential enrichment technology[17–19] (Supplementary Fig. 7a). Four fluorescein isothiocyanate aptamers were incubated with endothelial cells to test their binding ability (Supplementary Fig. 7b,c). One candidate aptamer with the highest binding affinity was chosen for conjugating to agomiR-195. Flow cytometry analysis showed that aptamer candidates 2 (CA2) had higher binding ability to endothelial cells and satisfactory secondary structure (Supplementary Fig. 7b,c). Therefore, we chose the CA2 aptamer for the following experiment.

The aptamer-agomiR-195 was injected via tail vein of 12-month-old WT mice once per week for 3 months. qRT–PCR confirmed that intravenous injection of aptamer-agomiR-195 significantly increased the levels of miR-195 in ECs (Supplementary Fig. 7d). The number of CD31$^{hi}$Emcn$^{hi}$ endothelial cells in bone was significantly higher in aptamer-agomiR-195-treated mice than in vehicle-treated mice (Fig. 7a). Flow cytometric analysis showed a twofold increase in the number of CD31$^{hi}$Emcn$^{hi}$ endothelial cells in bone marrow of aptamer-agomiR-195–treated mice (Fig. 7c,d). However the total endothelial cells number was not changed (Fig. 7e). The trabecular bone volume and number, trabecular bone thickness was higher, but the trabecular separation was lower

after aptamer-agomiR-195-treatment (Fig. 7f–j). Aptamer-agomiR-195-treated mice also had higher number of osteoblasts and Osterix$^{+}$ osteoprogenitors on the bone surfaces (Fig. 7a,b and Supplementary Fig. 8a,b) while the number and surface of osteoclasts were not changed (Supplementary Fig. 8c–e). Calcein double labelling confirmed that trabecular BFR increased in mice treated with aptamer-agomiR-195 (Fig. 7k–m). Furthermore, bone strength was increased in those mice received aptamer-agomiR-195 treatment (Fig. 7n,o).

Taken together, these results suggested that EC-specific activation of miR-195 by intravenous injection of aptamer-agomiR-195 promoted the CD31$^{hi}$Emcn$^{hi}$ vessel formation, and stimulated new bone formation in aged mice.

## Discussion

Type H vessels mediate growth of the bone vasculature, generate distinct metabolic and molecular microenvironments, maintain perivascular osteoprogenitors and couple angiogenesis to osteogenesis[5,6,20]. The mechanism of type H vessel formation and degeneration is still unclear. Multiple miRNAs are important regulators of bone formation and resorption at the post-transcriptional level. However, no miRNAs have yet been identified that contribute to the regulation of type H vessel.

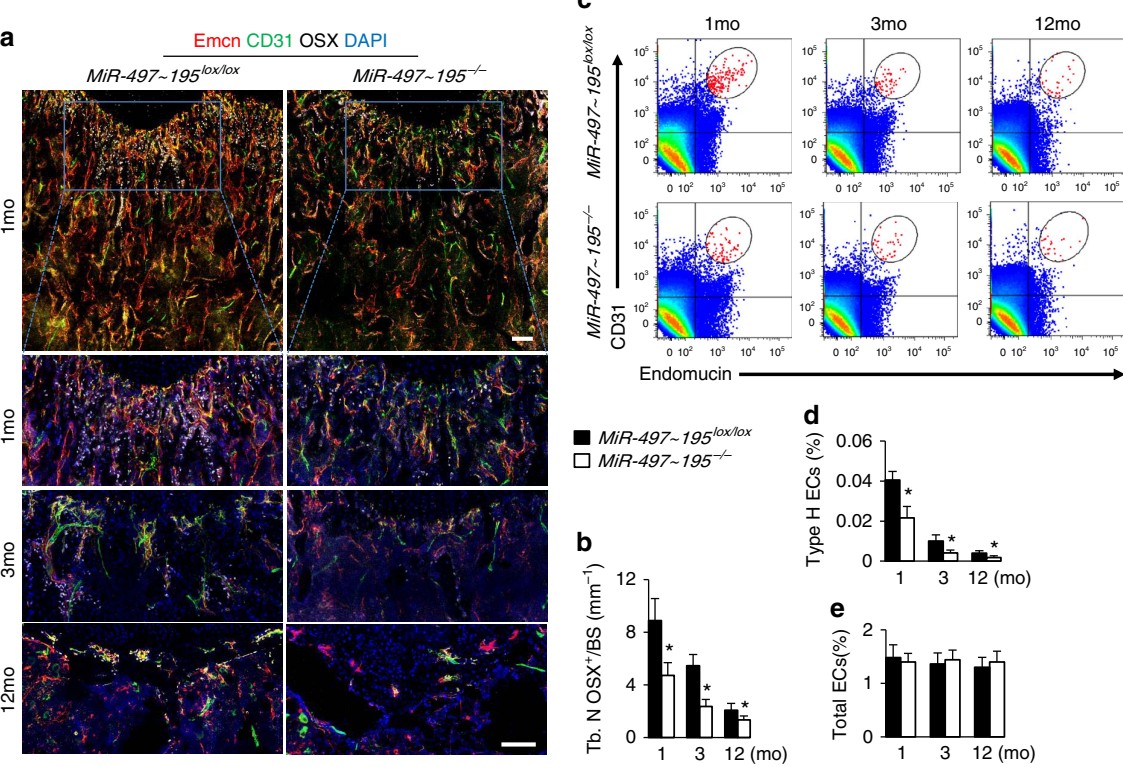

**Figure 3 | Endothelial-cell-specific MiR-497~195 cluster knockout mice showed less CD31^hiEmcn^hi vessels formation.** (**a**) Representative images of CD31 (green), Emcn (red) and Osterix (white) immunostaining in femora from 1-, 3- and 12-month-old *MiR-497~195^lox/lox* and miR-497~195 knockout (*miR-497~195^-/-*) mice. Scale bar, 100 μm. (**b**) Quantification of number of Osterix$^+$ osteoprogenitors in distal femora. ($n = 5$ mice in each group from three independent experiments). (**c–e**) FACS analysis dot plot (**c**) and quantitation of CD31^hiEmcn^hi endothelial cells (Type H ECs) (**d**) and total endothelial cells (Total ECs) (**e**) from long bone of the indicated age group. ($n = 6$ mice in each group from three independent experiments). Data shown as mean ± s.d. *$P < 0.05$, (Student's *t*-test).

In the present study, we found that the miR-497~195 cluster, which is strongly expressed in CD31^hiEmcn^hi endothelial cells and gradually decreased during ageing, plays an important part in type H vessel formation. miR-497 and miR-195 belong to miR-15 family, It has been reported that miR-497~195 negatively regulates angiogenesis in tumours[21–24]. Almeida *et al.* demonstrated that miR-195 in human primary mesenchymal stromal/stem cells negatively regulated proliferation, osteogenesis and had paracrine effect on angiogenesis by targeting vascular endothelial growth factor (VEGF)[25]. However, our results show that the miR-497~195 cluster is strongly expressed in CD31^hiEmcn^hi endothelial cells that promote angiogenesis and osteogenesis by maintaining endothelial Notch and HIF-1α activity in murine long bone. The discrepancy between our model system and the studies using hepatocellular carcinoma cells and prostate cancer cells is likely attributable to the distinct origin of the cells and microenvironments in which the cells reside. For instance, although Notch signalling inhibits vessel growth and endothelial proliferation in other organs, the same signalling promotes endothelial cell proliferation and angiogenesis in murine long bone[5,6]. The underlying mechanism of the distinct interaction of vessels and microenvironment in bone marrow and other organs merits further investigation.

Grünhagen *et al.*[26] showed that miR-497~195 cluster regulates osteoblast differentiation by targeting BMP signalling. We focused on the role of in endothelial cells in bone marrow. First, we revealed the role of the miR-497~195 cluster in the formation of type H vessel. Mice depleted of miR-497~195 in endothelial cells show significantly fewer CD31^hiEmcn^hi vessels, and lower bone mass compared to WT mice. Conversely, mice transgenically overexpressing miR-497~195 in endothelial cells showed significantly higher bone mass and more CD31^hiEmcn^hi vessels. We found that miR-497-195 increased CD31 expression in endothelial cells. It has been reported that treating the endothelial cells with PHD inhibitors, such as deferoxamine mesylate (DFM), would enhance HIF-1α stability and activity, and increase transcription of CD31 (ref. 5). In this study, we demonstrate that miR-497-195 directly target P4HTM, one member of the PHD family, to enhance HIF-1α stability. Thus, there is a possibility that miR-497-195 increase CD31 expression by inhibiting P4HTM to maintain HIF-1α stability. However, we cannot exclude the possibility that other mechanism of miR-497-195-mediated increase in the endothelial CD31 transcripts may exist. The underlying mechanism of miR-497-195 regulation of CD31 transcripts in the endothelium merits further investigation.

miRNAs mediate post-transcriptional gene silencing by base pairing to the complementary sites in the 3′-UTR of the target mRNA. We investigated the target genes by miR-497~195 in endothelial cells. It has been identified that endothelial HIF-1α and Notch activity influence type H vessel abundance and bone angiogenesis[5,6]. We found that miR-497~195 promoted Notch activity and HIF-1α protein expression by targeting Fbxw7 and P4HTM genes. Fbxw7 mediates polyubiquitination and proteasomal degradation of active Notch[13]. Ramasamy *et al.*[6] generated EC-specific Fbxw7 deficient mice that were characterized by an overactivation of Notch in endothelial cells. Type H vessel abundance and bone formation was increased in EC-specific Fbxw7 deficient mice. P4HTM promoted the degradation of HIF-1α under normoxic conditions[14]. P4HTM inhibitors, such as DFM, enhance HIF-1α stability and activity[27].

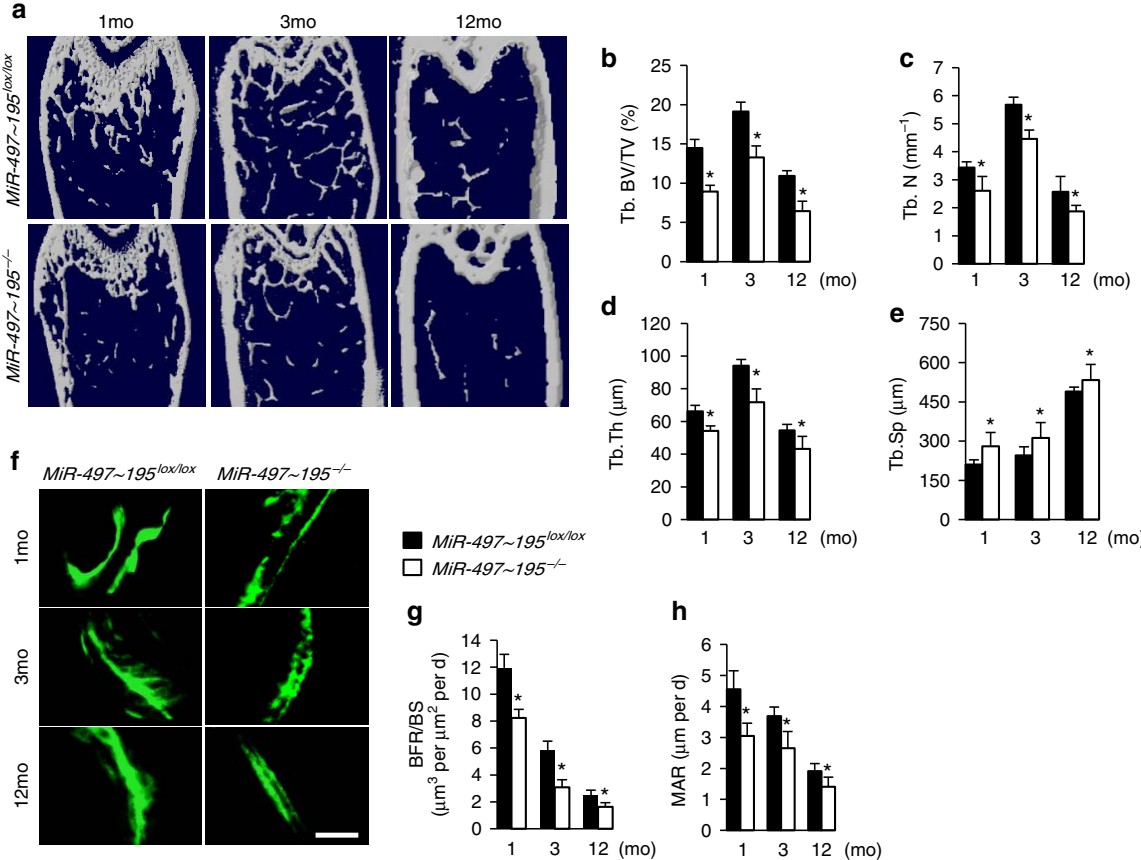

**Figure 4 | Endothelial-cell-specific MiR-497~195 cluster knockout mice showed less bone formation.** (**a–e**) Representative microcomputed tomography (μCT) images (**a**) and quantitative μCT analysis (**b–e**) of trabecular bone microarchitecture in femora from 1-, 3- and 12-month-old *MiR-497~195^lox/lox* and *miR-497~195^−/−* mice. (*n* = 5 mice in each group from three independent experiments). (**f–h**) Representative images of calcein double labelling of trabecular bone (**f**) with quantification of BFR per bone surface (BFR/BS) (**g**) and mineral apposition rate (MAR) (**h**) in femora of 1-, 3- and 12-month-old *MiR-497~195^lox/lox* and *miR-497~195^−/−* mice. (Scale bar, 25 μm. *n* = 5 mice in each group from three independent experiments). Data shown as mean ± s.d. *$P < 0.05$, (Student's *t*-test).

Kusumbe *et al.*[5] demonstrated that DFM administration led to substantial expansion of type H vessel endothelium and increased osteogenesis in aged mice. Our study showed that EC-specific miR-497~195 activation post transcriptionally inhibited Fbxw7 and P4HTM expression, maintained Notch activity and HIF-1α protein stability in endothelial cells and promoted type H vessel formation.

Aptamers are single-stranded nucleic acid molecules that bind to targets via folding into a three-dimensional structure with high affinity and selectivity[28]. Cell type-specific aptamers as drug delivery vehicles are currently undergoing clinical evaluation for ocular and haematological diseases and cancer[29,30]. In our study, we constructed EC-specific aptamers with secondary structures that are capable of molecular recognition specifically of endothelial cells, and combined them with angomiR-195 to specific elevate miR-195 level in endothelial cells. Our results showed that EC-specific activation of miR-195 increased type H vessel number and stimulates new bone formation in aged mice. As ageing is a major risk factor for osteoporosis, one of the major requirements for the treatment of age-related bone loss is to identify anabolic agents that can increase bone formation. Our results indicate that aptamer-mediated activation of miR-195 expression in endothelial cells promoted type H vessels formation and osteogenesis in aged mice, suggesting a novel strategy to treat age-related bone loss and senile osteoporosis. Together, our study provides mechanistic insight into how miRNA regulates angiogenesis during coupling with osteogenesis, while

also providing a further rationale for its therapeutic targeting to treat age-related osteoporosis.

## Methods

**Mice.** All animal care protocols and experiments were reviewed and approved by the Animal Care and Use Committees of the Laboratory Animal Research Center at Xiangya Medical School of Central South University. All mice were maintained in the specific pathogen-free facility of the Laboratory Animal Research Center at Central South University.

To generate endothelium specific miR-497~195 knockout mice, mice carrying loxP-flanked miR-497~195 alleles (*miR-497~195^lox/lox*) and Cdh5-Cre transgenics were interbred. The *miR-497~195^lox/lox* littermates were used as controls. The Cdh5-Cre transgenic mice (Stock No. 017968) and loxP-flanked miR-497~195 mice (Stock No. 34659-JAX) were purchased from Jackson Laboratory. For endothelium specific miR-497~195 knockout experiment, five to six male mice were used for each group at each observed time point (1, 3 and 12 months) for each independent experiment.

To generate endothelial-cell-specific miR-497~195 transgenic (Tg) mice, we first constructed Cdh5 pre-miR-497~195 vector by subcloned the mouse pre-miR-497~195 cDNA (synthesized by Genscript Co.) into the SalI-EcoRI site in a plasmid containing the Cdh5 promoter. Then the plasmid (Cdh5 pre-miR-497~195) was transfected into ECs using Lipofectamine 2000 (Invitrogen). Empty vector was used as control. We used qRT-PCR to detect the expression of miR-497 and miR-195. The fragments of the Cdh5 pre-miR-497~195 were purified and microinjected into C57BL/6 F2 mouse oocytes, and the oocytes were then surgically transferred into pseudopregnant C57BL/6 dams. One line with a highest (sixfold) overexpression of miR-497~195 in endothelial cells was selected from five transgenic founders and bred in C57BL/6 strain for six generations to obtain offsprings with a defined genetic background. The WT (C57BL/6) mice were used as controls. For endothelial-cell-specific miR-497~195 transgene experiment, five to six male mice were used for each group at each observed time point (1, 3 and 12 months) for each independent experiment.

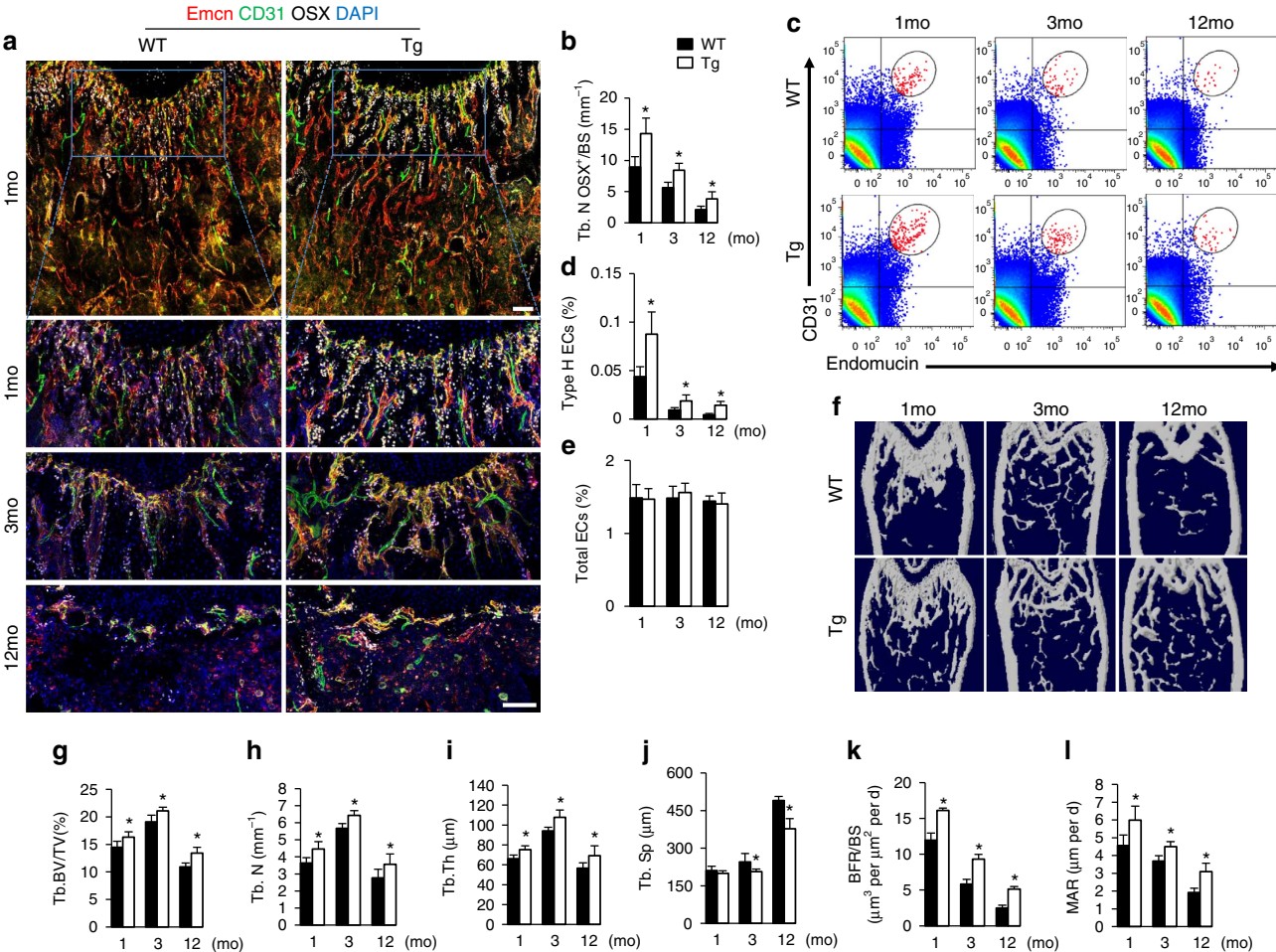

**Figure 5 | MiR-497∼195 cluster transgenic mice showed increased CD31[hi]Emcn[hi] vessels and bone formation.** (**a**) Representative images of CD31 (green), Emcn (red) and Osterix (white) immunostaining in femora from 1-, 3- and 12-month-old WT and miR-497∼195 transgenic (Tg) mice. Scale bar, 100 μm. (**b**) Quantification of number of Osterix+ osteoprogenitors in distal femora. (n = 5 mice in each group from three independent experiments). (**c–e**) FACS analysis dot plot (**c**) and quantitation of Type H ECs (**d**) and total endothelial cells (Total ECs) (**e**) from long bone of the indicated age group (n = 6 mice in each group from three independent experiments). (**f–j**) Representative microcomputed tomography (μCT) images (**f**) and quantitative μCT analysis (**g–j**) of trabecular bone microarchitecture in femora from 1-, 3- and 12-month-old WT and miR-497∼195 Tg mice. (n = 5 mice in each group from three independent experiments). (**k,l**) quantification of BFR per bone surface (BFR/BS) (**k**) and mineral apposition rate (MAR) (**l**) in femora of 1-, 3- and 12-month-old WT and miR-497∼195 Tg mice. (n = 5 mice in each group from three independent experiments). Data shown as mean ± s.d. *P < 0.05, (Student's t-test).

**MiRNA microarray assay.** Small RNAs were isolated from the total RNA of CD31[hi]Emcn[hi] cells and CD31[lo]Emcn[lo] cells from 1-month-old female C57BL/6 mice, and then labelled with Cy3. The OebiotechCompany performed the miRNA microarray assay. The fragmentation mixtures were hybridized to an Agilent-046065 Mouse miRNA Microarray V19.0 8 × 60 K (Agilent). Feature Extraction software 10.7.1.1 (Agilent) analysed the scanned images using default parameters to obtain background subtracted and spatially detrended processed signal intensities as the raw data. Raw data were normalized in a quantile algorithm with Genespring 12.0 (Agilent). Probes for which at least 100% of samples in any 1 condition out of 2 conditions had flags in 'Detected' were maintained. The raw data of the microarray have been uploaded to GEO with the accession number GSE95196.

**Flow cytometry.** For the analysis or sorting of CD31[hi]Emcn[hi] cells, CD31[lo]Emcn[lo] cells and total ECs in bone, we collected femora and tibiae after euthanization of WT mice or transgenic mice, removed epiphysis, muscles and periosteum around the bone, then crushed the metaphysis and diaphysis regions of the bone in ice-cold PBS to get the bone marrow. Then we used collagenase (Sigma, 11088793001, 2 mg ml$^{-1}$) to digest whole bone marrow at 37 °C for 20 min to obtain single-cell suspensions. After filtration and washing, the cells were counted and incubated for 45 min at 4 °C with endomucin antibody (Santa Cruz, SC-65495, 1:50), then washed and further incubated with APC-conjugated CD31 (R&DSystems, FAB3628A, 1:100), biotin-coupled CD45 (BD, 553077, 1:100) and Ter119 (BD, 559971, 1:100) antibodies for 45 min at 4 °C. We performed acquisition on a FACS FACScan cytometer (BD Immunocytometry Systems). For demarcating and sorting CD31[hi]Emcn[hi] cells, first standard quadrant gates were set, subsequently to

differentiate CD31[hi]Emcn[hi] cells from the total double positive cells in quadrant 2 gates were arbitrarily set at >10$^3$ log Fl-4 (CD31-APC) fluorescence and >10$^3$ log Fl-2 (endomucin-PE) fluorescence. CD31+CD45−Ter119− cells were then sorted according to side scatter and CD31-APC fluorescence at >10$^2$ log Fl-4 (CD31-APC) fluorescence after negative selection of leucocyte common antigen CD45 and Ter119 at <10$^2$ log Fl-1 (CD45-FITC) fluorescence and <10$^2$ log Fl-3 (Ter119-PerCP) fluorescence. We sorted CD31+CD45−Ter119− cells as total BMECs from total bone marrow cells.

**Culture of BMECs from mouse and human.** For mouse BMECs isolation, we collected bone marrow from tibiae and femurs of WT mice in sterile Ca$^{2+}$ and Mg$^{2+}$ free PBS. For human BMECs isolation, we collected bone marrow from 33 female patients with osteoarthritis or fracture undergoing knee joint replacement or open reduction internal fixation. After crushing the bone, the mixture was digested with collagenase A (Sigma, 2 mg ml$^{-1}$) to obtain a single-cell suspension. Endothelial cells were then sorted using endomucin antibody (cat. no. SC-65495, 1:50) and Dynabeads sheep anti-Rat IgG (Invitrogen). Sorted endothelial cells were then plated on dishes coated with fibronectin and cultured in endothelial cell growth medium (EBM-2, Clonetics; Lonza) supplemented with EGM-2 Single Quots (CC-4176, Clonetic; Lonza). At first passage, cells were again MACS sorted with endomucin antibody and plated for culture. Cells were fed every third day and passaged upon confluency. Cultures were maintained at 37C with 5% CO$_2$ in a humidified atmosphere. Mouse BMEC cultures between passage 2 and 5 were used for agomir-497∼195 or antagomir-497∼195 treatment and subsequent qPCR or western blot analysis.

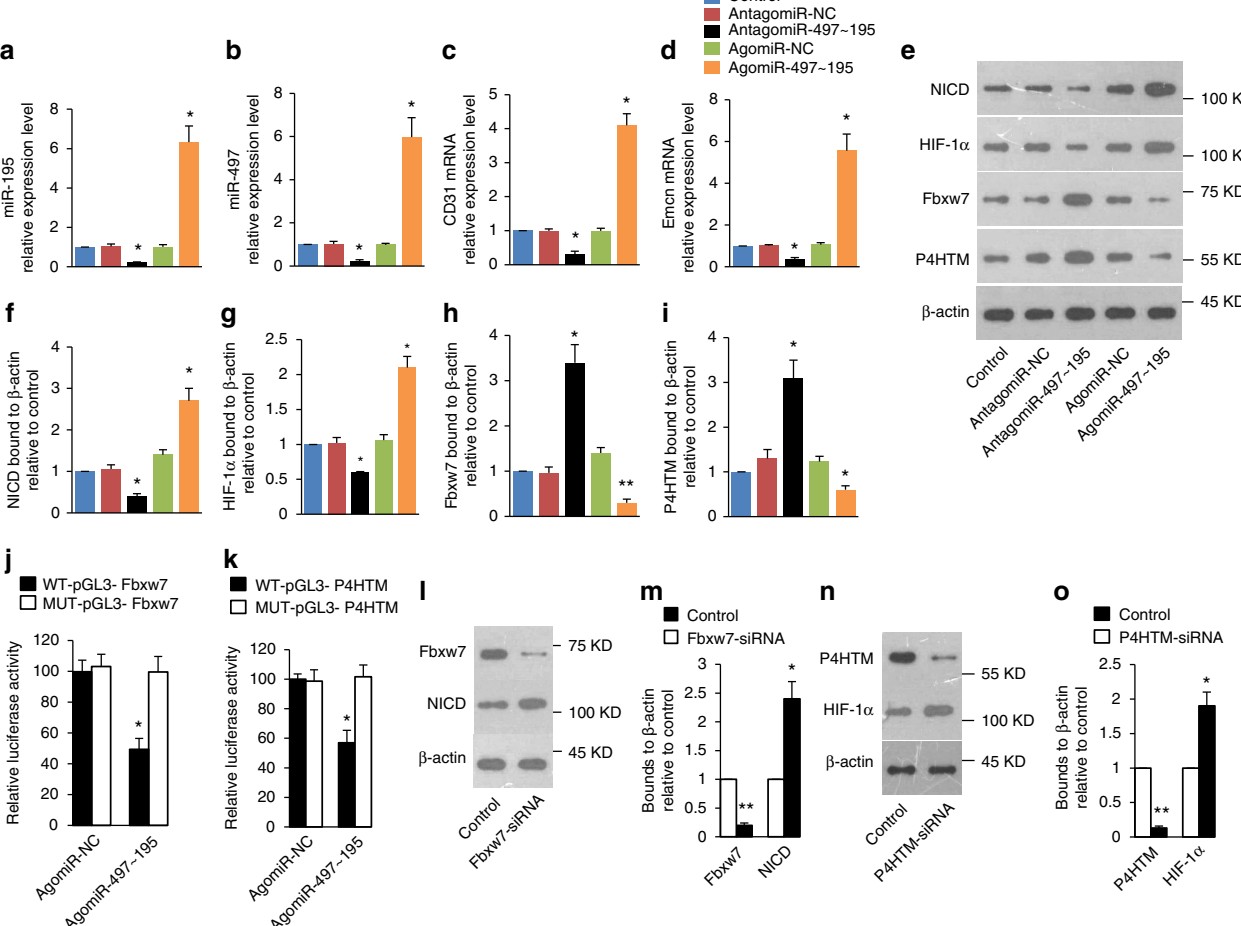

**Figure 6 | MiR-497~195 cluster targets Fbxw7 and P4HTM to maintain endothelial Notch and HIF-α activity. (a,b)** qRT-PCR analysis of the relative levels of miR-195 (**a**) and miR-497 (**b**) expression in BMECs transfected with agomiR-497~195, antagomiR-497~195 or their negative controls. NC, negative control. (**c,d**) qRT-PCR analysis of the relative levels of CD31 (**c**) and Emcn (**d**) mRNA expression in endothelial cells. (**e–i**) Western blot analysis (**e**) and the quantitation of the relative levels of NICD (**f**), HIFα (**g**), Fbxw7 (**h**) and P4HTM (**i**) protein expression in BMECs transfected with agomiR-497~195, antagomiR-497~195 or control. ($n = 3$ in each group from three independent experiments). (**j,k**) ECs were transfected with luciferase reporter carrying WT or MUT 3'-UTR of the Fbxw7 or P4HTM gene, WT-pGL3-Fbxw7 or MUT-pGL3-Fbxw7 (**j**) WT-pGL3-P4HTM and MUT-pGL3-P4HTM (**k**) respectively, and cotransfected with agomiR-497~195 or agomiR-NC. Effects of miR-497~195 on the reporter constructs were determined at 48 h after transfection. Firefly luciferase values, normalized for renilla luciferase, are presented. ($n = 3$ in each group from three independent experiments). (**l,m**) Western blot analysis (**l**) and the quantitation (**m**) of the relative levels of Fbxw7, NICD protein expression in BMECs transfected with Fbxw7 siRNA or control. (**n,o**) Western blot analysis (**n**) and the quantitation (**o**) of the relative levels of P4HTM, HIFα protein expression in BMECs transfected with P4HTM siRNA or control. ($n = 3$ in each group from three independent experiments). Data shown as mean ± s.d. *$P < 0.05$, **$P < 0.01$. (**j,k,m** and **o** Student's $t$-test; **a–d,f–i**, analysis of variance (ANOVA)).

**μCT analysis.** Femora were dissected from mice, fixed overnight in 10% formalin and analysed by high-resolution μCT (Skyscan 1172, Skyscan). The scanner was set at a voltage of 65 kV, a current of 154 μA and a resolution of 13.98 μm per pixel. The image reconstruction software (NRecon v1.6), data analysis software (CTAn v1.9) and three-dimensional model visualization software (μCTVol v2.0) were used to analyse the parameters of the distal femoral metaphyseal trabecular bone. We selected the region of interest for analysis as 5% of femoral length below the growth plate. Trabecular bone volume per tissue volume (Tb. BV/TV), trabecular number (Tb. N), trabecular separation (Tb. Sp) and trabecular thickness (Tb. Th) were measured.

**Immunohistochemistry and immunofluorescence staining.** Immunohistochemical staining was performed as described previously[31,32], Freshly femora were dissected from mice, fixed overnight at 4 °C in 10% formalin for 24 h, decalcified in 10% EDTA (pH 7.4) for 21 days and then embedded in paraffin. Four-micrometer-thick longitudinally oriented bone sections were used for staining. We used OCN and TRAP staining to quantify number and surface of osteoblasts, number and surface of osteoclasts, respectively. We counted the numbers of positively stained cells in four random visual fields of distal metaphysis per femur in five sequential sections per mouse in each group and normalized them to the number per millimetre of adjacent bone surface (N mm$^{-1}$) in trabecular bone.

For immunofluorescence staining freshly dissected bone tissues collected from transgenic mice and their control littermates were immediately fixed in ice-cold 4% paraformaldehyde solution for 4 h, and decalcified in 0.5 M EDTA (pH 7.4) at 4 °C for 24 h (1-month-old mice) or for 48 h (12 and 15 months old mice). The bones were then incubated in 20% sucrose and 2% polyvinylpyrrolidone (PVP) solution for 24 h, as described previously[33–35]. Finally, we embedded the tissues in 8% gelatin (porcine; Sigma, G2500) in presence of 20% sucrose and 2% PVP (Sigma, PVP360). Forty-micrometer-thick longitudinally oriented bone sections were stained with individual primary antibodies to mouse CD31 (Abcam, ab28364, 1:100), endomucin (Santa Cruz, V.7C7, 1:50) and Osterix (Abcam, ab22552, 1:100,), overnight at 4 °C. Subsequently, we used secondary antibodies conjugated with fluorescence (Jackson ImmunoResearch, 415-605-166, 1:500; 315-545-003, 1:500; 315-605-003, 1:250) at room temperature for 1 h while avoiding light. We used isotype-matched controls, such as polyclonal rabbit IgG (R&D Systems, AB-105-C), polyclonal goat IgG (R&D Systems, AB-108-C) and monoclonal rat IgG2A (R&D Systems, 54447) under the same concentrations and conditions as negative controls. The sections were observed under a confocal microscope (FLUOVIEW FV300, Olympus).

To examine dynamic bone formation, we subcutaneously injected 0.1% calcein (Sigma, 20 mg kg$^{-1}$ b.w.) in PBS into the mice 8 and 2 days before killing. We observed calcein double labelling in undecalcified bone slices under a fluorescence

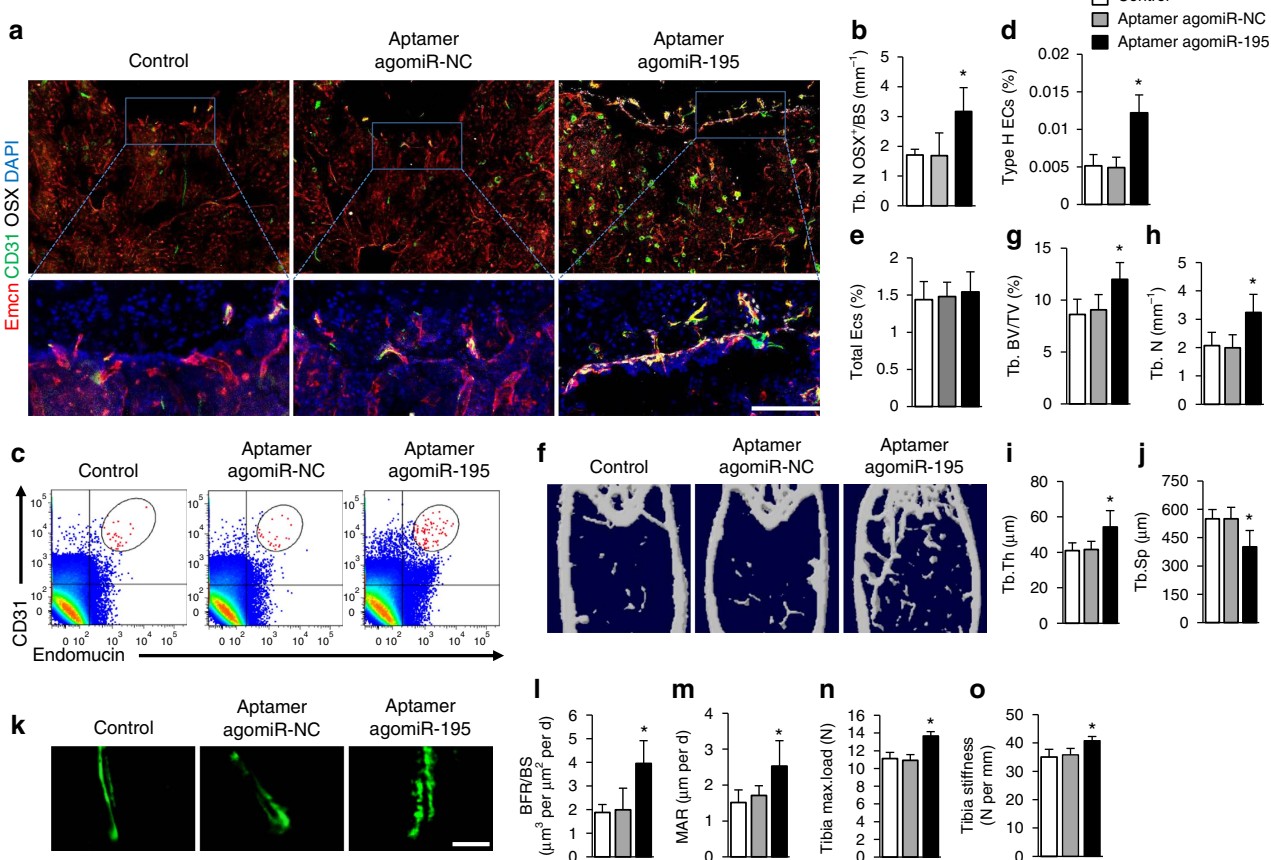

**Figure 7 | Injection of aptamer-agomiR-195 increased CD31^hiEmcn^hi vessel and bone formation.** Aptamer-angomiR-195 was injected via tail vein of 12-month-old mice once per week for 3 months. NC, negative control. (**a**) Representative images of CD31 (green), Emcn (red) and Osterix (white) immunostaining in femora from aptamer-agomiR-195 treated mice and their control group. Scale bar, 100 μm. (**b**) Quantification of number of Osterix^+ osteoprogenitors in distal femora. ($n = 5$ mice in each group from three independent experiments). (**c–e**) FACS analysis dot plot (**c**) and quantitation of CD31^hiEmcn^hi endothelial cells (Type H ECs) (**d**) and total endothelial cells (Total ECs) (**e**) from long bone of the aptamer-agomiR-195 treated mice. ($n = 6$ mice in each group from three independent experiments). (**f–j**) Representative microcomputed tomography (μCT) images (**f**) and quantitative μCT analysis (**g–j**) of trabecular bone microarchitecture in femora from aptamer-agomiR-195 treated mice and their control group. (**k–m**) Representative images of calcein double labelling of trabecular bone (**k**) with quantification of BFR per bone surface (BFR/BS) (**l**) and mineral apposition rate (MAR) (**m**). (Scale bar, 25 μm. $n = 5$ mice in each group from three independent experiments). (**n,o**) Three-point bending measurement of tibia maximum load. ($n = 5$ mice in each group from three independent experiments). Data shown as mean ± s.d. *$P < 0.05$, (analysis of variance (ANOVA)).

microscope. Four randomly selected visual fields in distal metaphysis of femur were measured to test trabecular bone formation.

**Three-point bending test.** To measure the cortical strength of the tibia, the mechanical-testing machine equipped with a 500 N M-SI sensor (Celtron Technologies Inc.) was used to run 3-point bending test. There are 2 end support points and 1 central loading point for the 3-point bending test. The length span between 2 support points was 60% of the total bone length. Each bone was loaded at a constant speed of 0.155 mm s$^{-1}$ until failure. The biomechanical measurement data were collected from the load-deformation curves. The maximum load (N) and stiffness (N/mm) were recorded.

**Construction of EC-specific aptamer delivery system.** The selection technology to generate aptamers is called systematic evolution of ligands by exponential enrichment (systematic evolution of ligands by exponential enrichment)[28,36–38]. Briefly, ssDNA library was incubated with $1–2 \times 10^6$ endothelial cells at 37 °C for 0.5 h. The bound DNAs were eluted and then incubated with non-endothelial cells at 37 °C for 0.5 h. The supernatant was desalted and amplified by PCR. The desired ssDNA was separated by streptavidin-coated magnetic beads (Promega). After multiple rounds of selection, the enriched ssDNA pool was cloned into TOP3 chemically competent *Escherichia coli* with the TA cloning kit (Invitrogen) and sequenced by Sangon, Shanghai, China. Binding assays of enriched ssDNA with the endothelial cells, mononuclear cells and osteoblasts were performed by a FACScan cytometer (BD Immunocytometry Systems). Secondary structures were predicted by RNAstructure 5.6 software. The EC-aptamer candidate 2 (CA2), a 33 bases

single strand DNA sequence (5′-CCC ACG TCT GCG CTT AGC TCC TGG GCC TGG ATG GGC-3′) were selected among three candidate aptamers for its higher binding ability and satisfactory secondary structure.

AgomiR-195 and the respective negative control were synthesized by RiboBio Co. We mixed 1 part by volume of a polyethyleneimine solution (100 μg ml$^{-1}$, pH 6.0) with 6 parts by volume of a 4.2-μM sodium citrate to form the polyethyleneimine-citrate core structure (nanocore). Then, we added 3 parts by volume of synthetic EC aptamers (50 nM) and agomiR-195 (1 μM) to the nanocore for 5 min of reaction to assemble the nanocomplex. 12-month-old male mice received either 40 μl of an agomiR-195 nanocomplex or a comparable volume of PBS via tail vein injection once per week for 12 weeks.

**mRNA 3′-UTR cloning and luciferase reporter assay.** For functional analysis of miR-497∼195, the segments of the mouse Fbxw7 and P4HTM 3′-UTR, including the predicted miR-497∼195–binding site, were PCR-amplified. The PCR products were purified and inserted into the XbaI-FseI site immediately downstream of the stop codon in the pGL3 control luciferase reporter vector (Promega Corp.), resulting in mouse WT-pGL3-Fbxw7 or WT-pGL3-P4HTM vectors. The Fbxw7 and P4HTM mutants for the miR-497∼195 seed regions were prepared using the QuikChange Site-Directed Mutagenesis Kit (Stratagene) to get mouse MUT-pGL3-Fbxw7 or MUT-pGL3-P4HTM. Mouse BEPs were transfected with either WT or mutant pGL3 construct, the pRL-TK renilla luciferase plasmid (Promega Corp.), and agomiR-497∼195 or agomiR-NC for 48 h using Lipofectamine 2000 (Invitrogen). The dual luciferase reporter assay system (Promega Corp.) was used to quantify luminescent signal using a luminometer (Glomax; Promega Corp.).

Each value from the firefly luciferase assay was normalized to the renilla luciferase value from the cotransfected phRL-null vector (Promega Corp.).

**qRT-PCR analysis.** We performed qRT-PCR using a Roche Molecular Light Cycler as previously described[39,40]. TRIzol reagent (Invitrogen) was used to isolate the total RNA from tissues or cultured cells, then we used 1 µg total RNA and SuperScript II (Invitrogen) to reverse transcription. Amplification reactions were set up in 25-µl reaction volumes containing SYBR Green PCR Master Mix (PE Applied Biosystems) and amplification primers. The primer sequences used for real-time PCR were listed in Supplementary Tables 1 and 2. A 1-µl volume of cDNA was used in each amplification reaction.

**Western blot.** We performed western blotting as previously described[39]. Cell lysates were prepared from BMECs transfected with agomiR-497~195, antagomiR-497~195, Fbxw7 siRNA, P4HTM siRNA or their negative controls in cell lyses buffer with 2% sodium dodecyl sulfate, 2 M urea, 10% glycerol, 10 Mm Tris-HCl (pH 6.8), 10 mM dithiothreitol and 1 mM phenylmethylsulfonyl fluoride. Total cell lysates were separated by SDS–polyacrylamide gel electrophoresis blotted on polyvinylidene difluoride membranes (Millipore). The membranes were incubated with specific antibodies to Fbxw7 (Abcam, ab109617, 1:1,000), P4HTM (Abcam, ab94626, 1:500), HIF-1α (Abcam, ab16066, 1:2,000) and NICD (Abcam, ab52301, 1:1,500) then reprobed with appropriate horseradish peroxidase-conjugated secondary antibodies (Santa Cruz, 1:10,000). Blots were developed using an ECL Kit (Santa Cruz), and exposed to x-ray films. The uncropped scans of blots are shown in Supplementary Figs 9 and 10.

**Study population.** Human bone marrow samples were obtained from 33 patients (female) based on the inclusion and exclusion criteria. Nine patients with osteoarthritis undergoing knee joint replacement, with ages ranging from 50 to 73 years; 10 patients with tibia fracture and 14 patients with femur shaft fracture undergoing Open Reduction Internal Fixation ranging in age from 20 to 53 years (human bone marrow collection were conducted by the Orthopedic Surgery Department at Xiangya Hospital of Central South University, Changsha, Human, China). All subjects were screened using a detailed questionnaire, disease history and physical examination. Subjects were excluded from the study if they had conditions affecting bone metabolism, including diseases of the kidney, liver, parathyroid, thyroid, hyperprolactinemia, oophorectomy, rheumatoid arthritis, ankylosing spondylitis, malabsorption syndromes, malignant tumours, haematologic diseases or previous pathological fractures within one year. If the subjects had received treatment with glucocorticoids, estrogens, thyroid hormone, parathyroid hormone, fluoride, bisphosphonate, calcitonin, thiazide diuretics, barbiturates and anti-seizure medication, they were also excluded. These participants were performed bone marrow collection during bone fracture surgery and joint replacement.

The clinical study was approved by the Ethnic Committee of Xiangya Hospital of Central South University, and written informed consents were obtained from all participants before bone marrow collection.

**Statistics.** Data are presented as mean ± s.d. For comparisons of 2 groups, two-tailed Student's *t*-test was used. For comparisons of multiple groups, one-way ANOVA was used. All experiments were repeated at least three times to guarantee reproducibility of findings, and representative experiments are shown. Differences were considered significant at $P < 0.05$. No randomization or blinding was used and no animals were excluded from analysis. Sample sizes were selected on the basis of previous experiments.

**Data availability.** The data that support the findings of this study are available within the article and Supplementary Files or available from the authors upon request. Microarray data that support the findings of this study have been deposited in the GEO database with the accession code: GSE95196. The Microarray data referenced during the study are available in a public repository from the https://www.ncbi.nlm.nih.gov/geo/.

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

## Acknowledgements

This work was supported by Grant No. 81520108008 from the Major International (Regional) Joint Research Project of China National Natural Scientific Foundation (NSFC), Grant No. U1301222 from the NSFC-Guangdong Joint Project, Grant No. 81570806 from the NSFC, Talent Grant of Xiangya Hospital.

## Author contributions

X.-H.L., M.Y. and C.-J.L. designed the experiments; M.Y., C.-J.L. and X.S. carried out most of the experiments; Q.G., Y.X., T.S., M.-L.T., H.P., Q.L.u., Q.Liu and H.-B.H. helped to collect the samples. T.-J.J., M.-X.L., M.W. and X.C. proofread the manuscript; X.-H.L. supervised the experiments, analysed results and wrote the manuscript.

## Additional information

**Competing interests:** The authors declare no competing financial interests.

