## [Peer Review File · Nature Communications]

Reviewers' comments:

Reviewer #1 (expert in miRs and angiogenesis)

Remarks to the Author:

This manuscript examines the role of miRNA 497-195 regulating angiogenesis and osteogenesis. Overall this is emerging area and an interesting study but somewhat predictable since there is ample evidence that miR-15 family and specifically miR-497 targets several pathways driving angiogenesis and miR 195 targets VEGF.

Specific comments:

1. Since miRs 497 and 195 are part of the miR-15 family, it is important to evaluate their presence in Type H or L EC in Fig 1 since previous papers have shown their importance in angiogenic signaling.
2. The EC specific KO mice or TG mice have not been characterized very well. What is the level of excision and expression of other miR-15 members? What about other angiogenic functions since miR 497-195 is not exclusively expressed in only Type H cells.
3. How does miR-497-195 reduce CD31 which is expressed in all EC?
4. The EC targeted aptamer is poorly characterized.

Reviewer #2 (expert in osteogenesis and angiogenesis)

Remarks to the Author:

In their manuscript entitled 'MiR-497~195 cluster regulates angiogenesis during coupling with osteogenesis via maintaining endothelial Notch and HIF-1 α activity', Yang et al. report that miRNAs of the miR-497~195 cluster are important regulators of Notch and HIF signaling in the bone endothelium. They show that the inactivation of the cluster specifically in endothelial cells (ECs) leads to loss of the Ecmhi CD31hi (so-called H-type) capillary subtype, which is associated with osteoprogenitors and has previously been shown to promote osteogenesis. The authors observe an age-related decline in miR-195 and miR-497 levels in murine and human ECs, and argue that this might contribute to the previously reported losses in H-type vessels and bone mass during ageing. Important roles of the miR-497~195 cluster are supported by EC-specific genetic inactivation and overexpression experiments. Based on in vitro experiments with cultured bone ECs, the authors argue that the miR cluster controls the levels of Fbxw7 and the prolyl 4-hydroxylase P4HTM, which regulate Notch and HIF signaling, respectively. Finally, it is shown that aptamer-mediated delivery of an agonistically acting agomiR-195 increased the number of H-type vessels and bone formation in aged mice.

This is a potentially interesting study, which addresses an important aspect of vessel-tissue crosstalk in the skeletal system. A number of weaknesses of the current version limit the strength of key conclusions, but it should be relatively straightforward to address these important issues.

P4HTM is a relatively little studied prolyl 4-hydroxylase (PHD). Knockout mice are viable and defects in bone or vasculature have not been reported yet. Thus, it is surprising that the role of the PHDs EglN1-3, which are much better established regulators of HIF signaling, have not been investigated. Which PHDs are expressed in the bone endothelium and is there functional redundancy with P4HTM? Are EglN1-3 controlled by the miR-497~195 cluster? This is a critical issue that needs clarification.

As the FACS strategy for the sorting of bone ECs is the basis for much of the data shown, dot plots for the isolation of hi/hi and lo/lo ECs should be included.

Improve the quality and increase the magnification of figures 2a, 3a, and 4a. Figures 2c, 3k and 4j lack vessel staining to support the association with OSX+ cells. Lower mag overview images and high magnification details should be provided.

What are the efficiencies of the miR-497/195 knockout and the level of transgenic overexpression expression in bone ECs in vivo? Are there alterations in the proliferation of bone ECs (best EdU+ or BrdU+ ECs by FACS) or artery formation?

Molecular weight markers are missing in all Western blots and should be indicated. Add quantitation of Western blot data. Is it correct that the in vitro analysis of HIF stability performed at room atmosphere?

The grammar of the manuscript should be improved.

Line 87 (Introduction): Ref. 6 is not the correct reference here. This paper has investigated the role of PDGF-B secretion by preosteoclasts and the analysis of Emcn^{hi} CD31^{hi} vessels was used as one of the readouts.

Line 257 (Discussion). Ref. 16 is about the first identification of HIF regulation by a prolyl 4-hydroxylase but actually does not refer to P4HTM. Koivunen et al. (JBC 2007) should be mentioned here.

We would like to thank the reviewers for their thoughtful and constructive comments regarding our manuscript. We have addressed all of the questions and comments brought forth through additional experimentation and clarification. We hope that the reviewers will find our responses to their comments satisfactory, and we are willing to finish the revised version of the manuscript including any further suggestion that the reviewers may have.

The following responses have been prepared to address the two reviewers' comments in a point-by-point fashion.

Reviewer #1 (expert in miRs and angiogenesis)

This manuscript examines the role of miRNA 497-195 regulating angiogenesis and osteogenesis. Overall this is emerging area and an interesting study but somewhat predictable since there is ample evidence that miR-15 family and specifically miR-497 targets several pathways driving angiogenesis and miR 195 targets VEGF.

Response:

We appreciate Reviewer #1's the constructive comments. We agree that several members of miR-15 family play important roles in regulating angiogenesis. It has been reported that miR-195-497 negatively regulate angiogenesis in multiple tumors^{1, 2, 3, 4}. MiR-497 post-transcriptionally down-regulates vascular endothelial growth factor A (VEGFA) in hepatocellular carcinoma (HCC) cells and pancreatic cancer cells^{2, 4}. Meanwhile, miR-195 directly inhibits the expression of VEGF and E-cadherin in HCC cells and prostate cancer cells^{1,3}.

Our project identified the miR-497~195 cluster was strongly expressed in CD31^{hi}Emcn^{hi} endothelial cells which would promote specific vessel subtype: type H vessel as well as osteogenesis by maintaining endothelial Notch and HIF-1 α activity in murine long bone. The discrepant results from our model system and the studies using HCC cells and prostate cancer cells are likely attributable to the distinct origin of the cells and microenvironments in which the cells reside. For instance, although Notch signaling inhibits vessel growth and endothelial proliferation in other organs, the signaling promotes endothelial cell proliferation and angiogenesis in murine long

bone⁵. The underlying mechanism of the distinct interaction of vessels and microenvironment in bone marrow and other organs is worthy for further study.

We have discussed this important point in the revised manuscript (page 9, line 11-25)

Specific comments:

1. Since miRs 497 and 195 are part of the miR-15 family, it is important to evaluate their presence in Type H or L EC in Fig 1 since previous papers have shown their importance in angiogenic signaling.

Response:

We agree with the reviewer that the levels of other members of the miR-15 family should be evaluated in Type H or L endothelial cells (ECs). We compared the expression of miR-15a, miR-15b and miR-16 between Type H and Type L ECs by performing the miRNA microarray analysis. The results show that the expression levels of miR-15a, miR-15b and miR-16 have slight difference between Type H and Type L ECs. We further validated this result by quantitative real-time PCR. The expression levels of miR-15a, miR-15b and miR-16 slightly increased in CD31^{hi}Emcn^{hi} endothelial cells related to CD31^{lo}Emcn^{lo} endothelial cells. Whereas, the elevation of miR-15a, miR-15b and miR-16 expression levels was much less significant than miR-497~195, indicating the unique role of miR-497~195 in CD31^{hi}Emcn^{hi} endothelial cells. The new data was added in Supplemental Figure 1a and described in Result section in the revised manuscript.(page4, line 13-17)

2. The EC specific KO mice or TG mice have not been characterized very well. What is the level of excision and expression of other miR-15 members? What about other angiogenic functions since miR-497~195 is not exclusively expressed in only Type H cells.

Response:

The reviewer is right, the description of EC specific miR-497~195KO mice and TG mice needs to be further characterized. Firstly, we measured the expression level of

miR-15a, miR-15b and miR-16 in the endothelial cells (ECs) specific miR-497~195KO mice and TG mice. The results show that genetically manipulation of miR-497~195in ECs have little effects on other members of miR-15 family. Then, we tested the angiogenic functions in ECs specific miR-497~195KO mice and TG mice. We found that the fraction of EdU⁺ ECs in bone marrow was significantly increased in miR-497~195 TG mice but dramatically decreased in miR-497~195 KO mice relative to their wild type control indicating the positive regulation of miR-497~195in proliferation of bone ECs. Furthermore, we tested the artery formation in EC specific miR-497~195KO mice and TG mice. The immunofluorescence staining using α -SMA antibody showed that the branch number of α -SMA⁺ artery had no significant difference in EC specific miR-497~195KO or TG mice compared with their wild type mice.

This result indicate that MiR-497~195 increase endothelium proliferation and promote angiogenesis at the distal end of the arterial network without influence on artery formation.

The new data about expression level of miR-15 members were added in Supplemental Figure 1a and described in Result section in the revised manuscript. (page5, line 2-5, line 26-28)

The new data about angiogenic functions in the EC specific KO mice or TG mice models were added in Supplemental Figure 2a-d and Supplemental Figure 4a-d. We described in Result section in the revised manuscript. (page5, line 8-12; page6, line 3-5)

3. How does miR-497-195 reduce CD31 which is expressed in all EC?

Response:

We are sorry for not interpreting our data accurately. In fact, we found that miR-497-195 increased CD31 expression in endothelial cells. However this is a very good suggestion to figure out the specific signaling of how miR-497-195 increasing

CD31 expression in endothelium. It has been reported that treating the endothelial cells with PHD inhibitors, such as deferoxamine mesylate (DFM), enhance HIF-1 α stability and activity, would increase transcription level of CD31 level⁵. In our project we demonstrated that miR-497-195 directly target P4HTM, one member of the PHD family, to enhance HIF-1 α stability. Thus, there is a possibility that miR-497-195 increase CD31 expression by inhibit P4HTM and maintain HIF-1 α stability. But we can't exclude the possibility that other mechanism of miR-497-195 increasing CD31 transcript in endothelium may also exist. In this project, we used CD31 along with endomucin as makers to valuate type H endothelium formation. We mainly focus on the role of miR-497-195 in promoting Type H vessel formation by maintaining the endothelial Notch activity and HIF-1 α stability (both signaling pathway had been reported with the function of promoting Type H vessel formation). The underlying mechanism of miR-497-195 direct and indirect regulation of CD31 transcript in endothelium is worthy for further study.

We have discussed this important point in the revised manuscript (page10, line 2-10)

4. The EC targeted aptamer is poorly characterized.

Response:

The reviewer is right, the description of EC targeted aptamer in the previous version is not clear. We have added more detailed description of the EC targeted aptamer in the revised manuscript. (page8, line 7-15; page10, line 25-30)

1. Wang R, *et al.* MicroRNA-195 Suppresses Angiogenesis and Metastasis of Hepatocellular Carcinoma by Inhibiting the Expression of VEGF, VAV2, and CDC42. *Hepatology* **58**, 642-653 (2013).
2. Yan JJ, *et al.* MiR-497 suppresses angiogenesis and metastasis of hepatocellular carcinoma by inhibiting VEGFA and AEG-1. *Oncotarget* **6**, 29527-29542 (2015).
3. Chao C, *et al.* miR-195 inhibits tumor progression by targeting RPS6KB1 in human prostate cancer. *Clinical Cancer Research* **21**, 4922 (2015).
4. Liu A, Huang C, Cai X, Xu J, Yang D. Twist promotes angiogenesis in pancreatic cancer by

targeting miR-497/VEGFA axis. *Oncotarget* **7**, 25801-25814 (2016).

5. Kusumbe AP, Ramasamy SK, Adams RH. Coupling of angiogenesis and osteogenesis by a specific vessel subtype in bone. *Nature* **507**, 323-328 (2014).

Reviewer #2 (expert in osteogenesis and angiogenesis)

In their manuscript entitled 'MiR-497~195 cluster regulates angiogenesis during coupling with osteogenesis via maintaining endothelial Notch and HIF-1 α activity', Yang et al. report that miRNAs of the miR-497~195 cluster are important regulators of Notch and HIF signaling in the bone endothelium. They show that the inactivation of the cluster specifically in endothelial cells (ECs) leads to loss of the Emcnhi CD31hi (so-called H-type) capillary subtype, which is associated with osteoprogenitors and has previously been shown to promote osteogenesis. The authors observe an age-related decline in miR-195 and miR-497 levels in murine and human ECs, and argue that this might contribute to the previously reported losses in H-type vessels and bone mass during ageing. Important roles of the miR-497~195 cluster are supported by EC-specific genetic inactivation and overexpression experiments. Based on in vitro experiments with cultured bone ECs, the authors argue that the miR cluster controls the levels of Fbxw7 and the prolyl 4-hydroxylase P4HTM, which regulate Notch and HIF signaling, respectively. Finally, it is shown that aptamer-mediated delivery of an agonistically acting agomiR-195 increased the number of H-type vessels and bone formation in aged mice.

This is a potentially interesting study, which addresses an important aspect of vessel-tissue crosstalk in the skeletal system. A number of weaknesses of the current version limit the strength of key conclusions, but it should be relatively straightforward to address these important issues.

Response:

We appreciate Reviewer #2's overall encouragement and constructive comments which would help us in depth to improve the quality of the paper.

1. P4HTM is a relatively little studied prolyl 4-hydroxylase (PHD). Knockout mice are viable and defects in bone or vasculature have not been reported yet. Thus, it is surprising that the role of the PHDs EglN1-3, which are much better established regulators of HIF signaling, have not been investigated. Which PHDs are expressed in the bone endothelium and is there functional redundancy with P4HTM? Are EglN1-3 controlled by the miR-497~195 cluster? This is a critical issue that needs clarification.

Response:

Thanks for the reviewer's important suggestion. We have tested the expression of other PHDs in bone marrow endothelial cells (BMECs). High expression of PHD1 and PHD2, but low expression of PHD3 in BMECs were observed by Western blot analysis. More importantly, we tested whether miR-497~195 regulate PHDS expression post-transcriptionally by measuring PHDs levels after transfection of BMECs with antagomiR-497~195 or agomiR-497~195. Western blot analysis showed that increasing and decreasing miR- 497~195 level didn't affect PHD1, PHD2 or PHD3 protein level indicating the unique role of miR- 497~195 in regulating P4HTM in BMECs.

We have added the new data in the supplemental figure 6e, 6f. and we described in Result section in the revised manuscript. (page7, line 11-14)

2. As the FACS strategy for the sorting of bone ECs is the basis for much of the data shown, dot plots for the isolation of hi/hi and lo/lo ECs should be included.

Response:

The reviewer is right. We have added dot plots images for all the FACS analysis of type H endothelial cells in Figures 2c, 3c, 5c, 7c in the revised manuscript.

3. Improve the quality and increase the magnification of figures 2a, 3a, and 4a. Figures 2c, 3k and 4j lack vessel staining to support the association with OSX⁺ cells.

Lower mag overview images and high magnification details should be provided.

Response:

We have replaced figures 2a, 3a, 4a and 7a with higher quality images which provide lower mag overview and high magnification of type H vessel staining. Meanwhile, we co-stained OSX⁺ cells with the type H vessel in figures 2a, 3a, 5a and 7a in the revised manuscript which show the association of osteoprogenitor with type H endothelial cells.

4. What are the efficiencies of the miR-497/195 knockout and the level of transgenic overexpression expression in bone ECs in vivo? Are there alterations in the proliferation of bone ECs (best EdU⁺ or BrdU⁺ ECs by FACS) or artery formation?

Response:

We appreciate this important point. We have checked the expression level of miR-497/195 in ECs specific miR-497~195knockout mice and transgenic mice. We observed an excision rate of approximately 70% of the miR-497-195 alleles in miR-497-195 knockout mice and almost 6 times higher expression of miR-497-195 in transgenic mice relative to their own control. We have added these data in Supplemental Figure 1 b, d.

As the reviewer suggested, we tested the angiogenic functions in ECs specific miR-497~195KO mice and TG mice. We found that the fraction of EdU⁺ ECs in bone marrow was significantly increased in miR-497-195 TG mice but dramatically decreased in miR-497-195 KO mice relative to their wild type control. We also tested the artery formation in EC specific miR-497~195KO mice and TG mice. The immunofluorescence staining using α -SMA antibody showed that the branch number of α -SMA⁺ artery had no significant difference in EC specific miR-497~195KO or TG mice compared with their respective control. These result indicated that MiR-497~195 increase endothelium proliferation and promote type H endothelial cells in bone marrow without affecting artery formation. Similar questions were also raised by Reviewer #1 (please also see the response to Question #2 of Reviewer #1).

The new data about angiogenic functions in the EC specific KO mice or TG mice

models were added in Supplemental Figure 2a-d and Supplemental Figure 4a-d. We described in Result section in the revised manuscript. (page5, line 8-12; page6, line 3-5)

5. Molecular weight markers are missing in all Western blots and should be indicated. Add quantitation of Western blot data. Is it correct that was the in vitro analysis of HIF stability performed at room atmosphere?

Response:

We apologize for the unclear labeling and description of Western blots data in the original submission. We have labeled all Western blots data with molecular weight markers and added quantitation of all the Western blot data in Figure 6e-6i, 6l-6o, Supplemental Figure 6e and 6f in the revised version.

Yes, the in vitro analysis of HIF stability was performed at room atmosphere.

6. The grammar of the manuscript should be improved.

Response:

We have thoroughly revised the manuscript and carefully corrected the grammatical and typographical errors.

7. Line 87 (Introduction): Ref. 6 is not the correct reference here. This paper has investigated the role of PDGF-B secretion by preosteoclasts and the analysis of Emcnhi CD31hi vessels was used as one of the readouts.

Response:

We agree with the reviewer that Ref. 6 is not the correct reference regarding to the ability of type H vessel for mediating perivascular osteoprogenitors differentiation and coupling angiogenesis to osteogenesis. We have replaced it with the study of Kusumbe et al. (Nature 2014) .

8. Line 257 (Discussion). Ref. 16 is about the first identification of HIF regulation by a prolyl 4-hydroxylase but actually does not refer to P4HTM. Koivunen et al. (JBC

2007) should be mentioned here.

Response:

We agree with the reviewer that Ref. 16 is not the correct reference regarding to first identification of HIF-1 α regulation by P4HTM. We have replaced it with study of Koivunen et al (JBC 2007) in revised manuscript.

REVIEWERS' COMMENTS:

Reviewer #1 (Remarks to the Author):

The authors have addressed my concerns.

Reviewer #2 (Remarks to the Author):

All my questions and criticisms have been fully addressed in the revised manuscript.

There is just one minor point that requires attention: Suppl. Fig. 6a is supposed to show the expression of active Notch (NICD). This is impossible because active Notch is a proteolytic cleavage product of the full length receptor. qRT-PCR analysis could only investigate expression of the latter. In addition, there are 4 Notch receptors (Notch1-4) and it should be clarified which of those (presumably Notch1?) has been analyzed.

We would like to thank the reviewers for their overall encouraging comments regarding our revised manuscript. The detailed responses to the reviewer #2 are described below.

The following is a point-to-point response to the two reviewer's comments.

Reviewer #1 (Remarks to the Author):

The authors have addressed my concerns.

Response:

We are glad to hear that our revised version addressed Reviewer #1's concerns.

Reviewer #2 (Remarks to the Author):

All my questions and criticisms have been fully addressed in the revised manuscript.

There is just one minor point that requires attention: Suppl. Fig. 6a is supposed to show the expression of active Notch (NICD). This is impossible because active Notch is a proteolytic cleavage product of the full length receptor. qRT-PCR analysis could only investigate expression of the latter.

In addition, there are 4 Notch receptors (Notch1-4) and it should be clarified which of those (presumably Notch1?) has been analyzed.

Response:

We appreciate the encouraging comments and thoughtful suggestions on our revised manuscript.

We apologize for the unclear description and inaccurate interpretation regarding the qPCR analysis of Notch1 mRNA level in Supplementary Figure 6a. The reviewer is right, it is inappropriate to interpret the mRNA level of full length Notch receptor 1 as the activity of Notch signaling. Actually, we have detected the intracellular domain of Notch1 (NICD) protein levels in endothelial cells to measure the active Notch signaling by Western blot analysis (Fig. 6e). We also detected the Notch1 mRNA levels by qRT-PCR analysis (Suppl. Fig. 6a). We found that miR-497~195 regulated the production of NICD protein without changing the Notch1 mRNA levels in endothelial cells (Fig. 6e and Suppl. Fig. 6a). We have rephrase the description and interpretation of the result in the revised manuscript (Line 3-10, Page7).